# Direct Detection of Glutathione Biosynthesis, Conjugation, Depletion and Recovery in Intact Hepatoma Cells

**DOI:** 10.3390/ijms23094733

**Published:** 2022-04-25

**Authors:** Rex E. Jeffries, Shawn M. Gomez, Jeffrey M. Macdonald, Michael P. Gamcsik

**Affiliations:** Joint Department of Biomedical Engineering, Chapel Hill and North Carolina State University, University of North Carolina, Raleigh, NC 27695, USA; rex.jeffries@ncssm.edu (R.E.J.); smgomez@unc.edu (S.M.G.)

**Keywords:** glutathione, antioxidant, nuclear magnetic resonance spectroscopy, noninvasive, alkylating agent, metabolism

## Abstract

Nuclear magnetic resonance (NMR) spectroscopy was used to monitor glutathione metabolism in alginate-encapsulated JM-1 hepatoma cells perfused with growth media containing [3,3′-^13^C_2_]-cystine. After 20 h of perfusion with labeled medium, the ^13^C NMR spectrum is dominated by the signal from the ^13^C-labeled glutathione. Once ^13^C-labeled, the high intensity of the glutathione resonance allows the acquisition of subsequent spectra in 1.2 min intervals. At this temporal resolution, the detailed kinetics of glutathione metabolism can be monitored as the thiol alkylating agent monobromobimane (mBBr) is added to the perfusate. The addition of a bolus dose of mBBr results in rapid diminution of the resonance for ^13^C-labeled glutathione due to a loss of this metabolite through alkylation by mBBr. As the glutathione resonance decreases, a new resonance due to the production of intracellular glutathione-bimane conjugate is detectable. After clearance of the mBBr dose from the cells, intracellular glutathione repletion is then observed by a restoration of the ^13^C-glutathione signal along with wash-out of the conjugate. These data demonstrate that standard NMR techniques can directly monitor intracellular processes such as glutathione depletion with a time resolution of approximately < 2 min.

## 1. Introduction

Glutathione (GSH) is a tripeptide synthesized in two steps from the component amino acids, glutamate (glu), cysteine (cys) and glycine (gly) (Figure 1) and is present at relatively high concentration in most mammalian tissue. Glutathione is the most abundant low molecular weight thiol and acts an antioxidant, protecting cells from oxidative stress through a reduction in reactive oxygen/nitrogen species [1]. Glutathione may also protect cells by inactivating toxic electrophilic xenobiotics through direct conjugation or act to shield sensitive protein residues by a reversible formation of disulfide linkages [1]. As a key player in a myriad of cellular defense mechanisms, glutathione levels are used as a measure of tissue or organism-wide stress [2]; however, most studies measure tissue levels of glutathione or its oxidized counterpart, glutathione disulfide (GSSG, Figure 1), as a measure of redox balance in the cell [3]. Although the levels of these metabolites provide valuable information of the reduction potential, they do not reflect the capacity of a tissue to meet applied stresses. This capacity includes the ability of the tissue to replenish its glutathione stores in addition to the steady-state levels of the metabolite. This is illustrated by our work showing that glutathione steady-state levels are not related to drug response in breast cancer cells, but in all cases drug-resistant cells have an upregulated glutathione synthetic capacity [4,5]. This emphasizes the need to define both the metabolic profile of a tissue, i.e., its metabolome, along with dynamic information on metabolic flux, its ‘fluxome’.

Typically, time-dependent data on drug levels or their metabolic effects are determined through detailed pharmaco- and toxico-dynamics studies. Conventional approaches to determine the pharmaco- or toxico-kinetics of drugs consists of the use of body fluids (urine, plasma) and tissue to obtain drug and metabolite concentrations. This often requires samples to be collected at set time intervals that may miss rapid processes. If continued harvesting of tissue samples is necessary, this often precludes human studies and limits detailed analyses to animal models and still limits the number of sampling points that can be collected. In vivo noninvasive detection of drug and metabolite dynamics are available through a number of techniques including nuclear magnetic resonance (NMR) spectroscopy. In particular, ^13^C NMR enables the direct monitoring of multiple metabolic pathways and collection of pharmacodynamic data, mainly of glucose metabolism, noninvasively in intact tissue (e.g., [6,7]). Our laboratories have probed the use of ^13^C NMR to study glutathione metabolism in cultured cells [4,5] and in vivo in tumors [8,9], liver [10] and brain [11] by the administration of ^13^C-labeled substrates. Glutathione is readily amenable to such NMR studies due to its high concentration in cells and tissues. In cell studies, the control of administered substrates in cell growth media allows high isotopic enrichment thereby increasing sensitivity. In this report, we demonstrate this sensitivity by showing that the high fractional enrichment of glutathione achievable in cell culture enables rapid (1.2 min) acquisition of ^13^C NMR spectra. This temporal resolution allows for direct monitoring of the depletion of glutathione through conjugation reaction with the thiol-specific reagent monobromobimane (mBBr). In addition, the rate of formation of the bimane-glutathione conjugate and its export from the cell can be directly detected. These data show how magnetic resonance can provide detailed pharmacodynamic information on glutathione metabolism noninvasively in perfused cells.

## 2. Results

### 2.1. Cell Viability in the Fluidized Bed Bioreactor

Electrostatic encapsulation of JM-1 cells in calcium alginate by the methods presented resulted in solid beads of 500 μm in diameter that should not limit nutrient diffusion [12]. Perfusion of these beads in a fluidized bed bioreactor maintains cellular viability throughout the entire course of the experiments as determined by ^31^P NMR spectroscopy (Appendix A), showing high levels of nucleoside triphosphates (NTP, mainly ATP) that are characteristic of well-perfused cells [13].

### 2.2. ^13^C NMR Detection of Glutathione Synthesis and Depletion

The natural abundance ^13^C NMR spectrum of these cells shows resonances primarily due to glucose (glc) and other low molecular weight metabolites in the media (Figure 2A). The spectra in Figure 2 are each acquired with a total acquisition time of 30 min that was necessary to obtain good signal-to-noise ratios (S/N) on peaks of low intensities. Upon switching to media containing [3,3′-^13^C_2_]-cystine (^13^C-DMEM) a signal due to the labeled cystine is detectable at 39.1 ppm (Figure 2B). This spectrum also shows the addition of another new peak due to ^13^C label being incorporated into the cysteinyl-residue of glutathione, (γ-glutamyl-[3-^13^C]-cysteinyl-glycine), termed ^13^C-glutathione (^13^C-GSH), at 25.8 ppm that grows in intensity over time (Figure 2C). No signal for [3-^13^C]-cysteine was detected in spectra acquired under the described conditions.

This is due to the low concentration of intracellular cysteine in these cell lines (Table 1).

The time-course of label incorporation into glutathione is shown by acquiring and processing the data every 30 min (Figure 2D). Only the region of the spectrum containing the ^13^C-enriched glutathione signal is shown in Figure 2D and is plotted in blue in this Figure and subsequent time-course spectra. The data in Figure 2D reflects the increase in ^13^C fractional enrichment in glutathione upon the change from unlabeled to ^13^C-labeled cystine in the media and not a change in the intracellular glutathione content. We observed no other additional ^13^C resonances in these spectra from ^13^C-cyst(e)ine incorporation into other cellular components which is likely due to their low concentration.

After 13.5 h of perfusion with ^13^C-DMEM, the intracellular signal of the ^13^C-glutathione is strong with S/N = 53:1 for a 30 min acquisition (Figure 3A). With this strong signal intensity, subsequent ^13^C NMR spectra could be acquired much more rapidly, and the spectrum shown in Figure 3B, and subsequent time-course spectra, were acquired with total acquisition times of 1.2 min. In this shorter time frame, the signal for ^13^C-glutathione is still easily detectable with S/N = 12:1 (Figure 3B). After switching to this more rapid data acquisition mode, a bolus dose of the thiol scavenging alkylating agent monobromobimane (mBBr) in dimethylsulfoxide (DMSO), is added to the media and is infused for 10 min. The addition of mBBr rapidly depletes the ^13^C-glutathione signal and a resonance due to the ^13^C-glutathione-bimane conjugate (GS-bim) is visible in the cells at 33.6 ppm (Figure 3C). In addition, a signal due to the natural abundance ^13^C resonance of DMSO, used as a delivery vehicle for mBBr, is detectable at 39.0 ppm.

The peaks in the shaded region between 25 to 40 ppm in Figure 3 are replotted in Figure 4 as a time-course showing the changes in the NMR spectra of the ^13^C-GSH (blue), ^13^C-glutathione-bimane conjugate (GS-bim; red) and DMSO (black) just before and after the addition of mBBr to the perfused cells. This Figure shows the spectra acquired in 1.2 min time blocks over a total time of 4 h. A few spectra showing only ^13^C-GSH resulting from perfusion with ^13^C-DMEM were collected and then this medium was supplemented with mBBr in DMSO and infused for 10 min as shown in the timeline at the top of Figure 4. The perfusing medium was then switched back to ^13^C-DMEM to wash-out unreacted mBBr for the duration of the experiment. The data in Figure 4 show that the ^13^C-GSH peak was rapidly depleted by the addition of mBBr as alkylation of the glutathione sulfhydryl results in the formation of the GSH-bim conjugate. During and shortly after infusion of mBBr, the ^13^C-GSH in the cell was depleted as the rate of conjugation exceeded the rate of glutathione re-synthesis. The glutathione signal then recovered as the mBBr reagent was washed out of the cells and the rate of glutathione re-synthesis began to exceed the rate of depletion. The time-course for the ^13^C-bim conjugate grew in intensity to a maximum approximately 15 min after the introduction of the mBBr, then slowly washed out of the cells. The thiol reagent mBBr itself was not detectable under the conditions of the experiment; however, mBBr was dissolved in DMSO and added to the perfusion medium and since both mBBr [14] and DMSO easily pass through the cell membrane, the DMSO signal can be used as a surrogate marker of mBBr delivery and washout. For clarity, the time-course for the DMSO resonance is plotted separately and shown at the bottom of Figure 4. After 4 h, the ^13^C-GSH resonance had recovered nearly to pre-dosing levels as resynthesis of this peptide had replenished the intracellular glutathione stores. The conjugate was no longer detectable intracellularly after approximately 3.5 h. No ^13^C signal due to oxidized glutathione disulfide was detectable under the experimental conditions indicating that the mBBr depletion did not generate significant amounts of this species. At the conclusion of the experiment, another ^31^P spectrum showed the cells were still viable with no significant change in NTP and little change in the glycerophosphorylcholine (GPC) or glycerophosphorylethanolamine (GPE) peaks (Appendix A).

### 2.3. Reproducibility

The data from one of three experiments showing the initial uptake of [3,3′-^13^C_2_]-cystine to produce ^13^C-GSH (i.e., Figure 2D) are shown in Figure 5A. This figure plots the integrated intensity of the ^13^C-GSH peak at 25.8 ppm as a function of time. Data from two other experiments are shown in Appendix A. An exponential curve fit to the data is shown by the line in Figure 5A. Fitting the data from all three experiments yields a synthesis rate of 0.159 ± 0.023 h^−1^. The time course for mBBr-induced depletion of the ^13^C-GSH peak in one of three experiments using 1.2 min acquisitions (i.e., Figure 4), are shown in Figure 5B. The data from the other two experiments are shown in Appendix A. Here the data is much noisier due to the lower signal-to-noise ratio obtained with the shorter acquisition. Since different cell numbers were used in each experiment, and flow characteristics of the fluidized bed varied the number of cells within the sensitive area of the probe, these data were normalized. All experiments detected an approximately 60% depletion in ^13^C-GSH signal upon introduction of the mBBr. All glutathione levels recovered to within 10% of the original level of ^13^C-GSH detected at the start of the experiments. The rate of glutathione resynthesis appeared much faster than the rates determined from the data in Figure 5A. Although the depletion and recovery kinetics are complex, an estimate of the recovery rate can be obtained if we fit an exponential equation to the recovery data and this is shown in Figure 5B. Fitting the data to all three recovery data sets yields a value of 1.09 ± 0.166 h^−1^ or >5-fold increase in rate.

### 2.4. Fractional Enrichment

Mass spectrometry analyses of the extracted glutathione after three cell perfusion experiments show a ^13^C-fractional enrichment of glutathione of 0.823 ± 0.031 (*n* = 3). The levels of glutathione, cysteine, and γ-glutamylcysteine found in the JM-1 cells at the conclusion of these experiments are shown in Table 1.

### 2.5. Assessment of Mercapturic Acid Pathway Activity

The extracellular medium of JM-1 cells cultured in monolayer and treated with mBBr did not show the presence of N-acetyl-L-cysteinyl-bimane by UPLC assay. This suggests the mercapturic acid excretion pathway was not significant in JM-1 cells for the removal of glutathione-bimane conjugates.

## 3. Discussion

The high concentration of glutathione found in tissue is thought to be due to its important role in protecting cells from the stresses induced by reactive oxygen/nitrogen species and electrophilic metabolites, drugs and toxins [15]. These stressors can deplete glutathione in cells due to oxidation, which may be readily reversible, or through conjugation that may not be easily reversed. In the latter case, glutathione conjugates are often removed from the cell intact or metabolized to the mercapturic acid derivative and secreted [16]. In some cases, efflux of drugs or toxins may occur concomitantly with glutathione transport, e.g., MRP-based mechanisms [17]. In all these cases, glutathione levels can change transiently or permanently, and these changes can dictate cellular fate [18]. Glutathione levels in the cell are a balance between its rate of synthesis and depletion and therefore cellular fate is dependent on the relative rates of glutathione turnover. In cancer cells, we have found that cell response to chemotherapeutic challenge is better correlated with glutathione metabolic rates rather than its steady-state levels [4,5]. These experiments used NMR spectroscopy to measure glutathione turnover in unstressed cells where glutathione levels and metabolic rates were primarily controlled by feedback inhibition of glutamate cysteine ligase (*GCL)*, the enzyme controlling the first, rate-limiting step [19] in glutathione synthesis (Figure 1). These experiments demonstrated that metabolic rates in unstressed cells previously exposed to chronic chemotherapy was higher than in chemotherapy-naive parental cell lines [5]; however, none of these experiments determined how the cells would respond to acute chemotherapeutic challenge. Therefore, this report used mBBr, a thiol-specific reagent, as a model for the chemotherapeutic alkylating agents. This reagent was chosen for the initial experiments for several reasons. Firstly, mBBr is relatively hydrophobic and rapidly crosses the cell membrane. Secondly, the reagent reacts directly with glutathione or via glutathione *S*-transferase catalysis in an irreversible reaction to form a thio-ether conjugate that is similar to those observed with anticancer alkylating agents [20,21]. The rapidity of the reaction between glutathione and alkylating agents makes NMR detection more difficult. Our earlier experiments using intact perfused cells tracked synthesis and degradation of glutathione with time-points obtained every 42 min. In the current experiments, pre-labeling of the glutathione pool allowed facile detection of the glutathione with relatively short acquisition times. The spectra displayed in Figure 3B,C and Figure 4, were acquired in 1.2 min, and is much faster than our previous studies of glutathione metabolism by ^13^C NMR [4] and our earlier kinetic analyses of glutathione conjugation with alkylating mustards studied in a buffer solution with a 10 min time resolution using ^1^H NMR [22]. Although these acquisition times of 1.2 min are much faster than normal for direct ^13^C NMR detection, they are still far slower than 1 sec acquisition times used for studies employing hyperpolarized substrates (e.g., [23]); however, hyperpolarization of the glutathione pool has not been demonstrated to date and due to the short lifetime of the hyperpolarized signal, it would not have been possible to use this method to track the dynamic process of glutathione depletion, conjugate formation and resynthesis demonstrated in this work.

The high sensitivity demonstrated in these experiments is due in part to using cystine-deficient media. Essentially all media cystine is available as [3,3’-^13^C_2_]-cystine allowing subsequent glutathione enrichment to high levels. The JM-1 cells used in these experiments are four times smaller in volume than the drug-resistant MCF-7hc cells used in our previous experiments but with similar glutathione content; 66.8 ± 20.5 fmol/cell in JM-1 (Table 1) compared to 64.8 ± 12.4 or fmol/cell found in MCF-7hc cells [5]. Far more cells were required in these experiments than in previous work with MCF-7hc cells (2 × 10^7^ cells) [4] to obtain an equivalent signal. In addition, increased cell numbers in these experiments were required due to the use of a fluidized bed bioreactor, compared to a packed bed type bioreactor used previously. The fluidized bed offers improved mass transport characteristics compared to a packed bed, but the drawback is that the cells circulate within the NMR tube, therefore, at any given time, only a fraction of the cells are within the detectable ‘sweet spot’ of the NMR coils. This likely leads to some noise in the data as the cells circulate within the NMR tube. The equivalent signal obtained here and in the previous studies of MCF-7hc cells [4] shows this method would be amenable to studies in other cell lines.

The data show that the [3,3’-^13^C_2_]-cystine is readily detectable in the media but the reduced form of this labeled substrate [3-^13^C]-cysteine is not. If present at significant concentration (>0.5 mM) the NMR signal of [3-^13^C]-cysteine should appear at 25.2 ppm, close to, but distinguishable from that of ^13^C-glutathione (25.8 ppm). Whether the primary mechanism for uptake for cyst(e)ine is as the reduced form cysteine or the oxidized form cystine has not been described for JM-1 cells. In hepatocytes, intracellular cysteine is provided primarily by the uptake of cysteine via the alanine–serine–cysteine (ASC) system with a minimal uptake of cystine via the x_c_^-^ pathway [24]. Hepatocytes also exhibit the ability to provide intracellular cysteine via the transsulfuration pathway [24]. Interestingly, cystine uptake via x_c_^-^ may be an adaptive response for cultured cells due to the predominance of cystine rather than cysteine in the culture media [25] and may be operative in the JM-1 cells which we have maintained in culture for years. The JM-1 hepatoma line has not been characterized in regard to cyst(e)ine uptake mechanisms but the HepG2 hepatoma line has been characterized and is shown to rely upon ASC-mediated cysteine uptake [26]. For the HepG2 line in cell culture, where cystine dominates in the media, glutathione reacts with the cystine to form a mixed disulfide liberating free cysteine for uptake into the cell [26]. Whether the JM-1 cell line relies exclusively upon the ASC system or whether the x_c_^-^ system is present is not known but many cancer cell lines do appear to contain at least some capacity for cystine uptake [25,27]. We have also determined that in unstressed JM-1 cells, the contribution of the transsulfuration pathway to cysteine production is negligible, which is consistent with what is observed in Hep G2 and HuH-7 hepatoma cells [28].

Acquiring good quality NMR data over multiple timepoints for the synthesis of glutathione (Figure 2) with a 30 min time resolution is not difficult as its synthesis rate is relatively slow; however, the reaction of glutathione with alkylating agents such as mBBr, via enzyme-catalyzed or non-catalyzed reactions occur much more quickly. The results in Figure 4 show that we have the sensitivity to follow relatively rapid glutathione conjugation reactions in intact cells. The data shows that efflux of the glutathione-bimane conjugate is relatively slow compared to its synthesis resulting in an initial buildup of conjugate within the cell followed by a slow efflux. The glutathione-bimane conjugate may be exported from the cell via the multi-drug resistance related protein (MRP) [29] but it is not known whether the JM-1 cell line contains MRP activity. In addition, we see no evidence in the NMR data that significant amounts of N-acetylcysteine-bimane, i.e., the expected mercapturic acid pathway metabolite, is present in the cells, especially in hepatocytes [30]. This would lead to a small but detectable shift in the ^13^C resonance if present at significant levels. In order to determine whether metabolism via the mercapturic pathway was present and metabolites did not accumulate to NMR-detectable levels, we collected perfusate after mBBr treatment and analyzed it by HPLC and found no indication of N-acetylcysteinyl-bimane. Similarly, in monolayer experiments we found no evidence for N-acetylcysteinyl-bimane in the media from mBBr-treated JM-1 cells, appearing to rule out significant mercapturic acid metabolic activity for efflux of the glutathione-bimane conjugate.

As the glutathione-bimane conjugate is formed, the glutathione levels drop and then begin to recover as the mBBr bolus is cleared and the synthetic rate exceeds the rate of depletion. Although the kinetics of label incorporation into ^13^C-GSH and depletion and recovery requires a more complex analysis, fitting exponential functions to the data suggests that the restoration of glutathione occurs at a rate that is approximately five-fold that found when initially labeling the glutathione pool. This is consistent with glutathione serving as a feedback inhibitor of glutamate cysteine ligase (*GCL*) as its depletion would release this enzyme for increased synthetic activity. In cell-free preparations, feedback inhibition at levels >K_i_ resulted in a lowering of synthesis rates by approximately one-third of the non-inhibited rate [19,31] and is consistent with our results. Interestingly, in intact yeast cells, the presence of feedback inhibition of *GCL* has been questioned [32]. Unlike studies in cell-free preparation, both the yeast studies and our studies were performed on intact cells that reflect the influence of substrates and co-factors at physiological levels. The different results therefore suggest that the glutathione system may be under different control mechanisms in yeast cells than those found in mammalian tissue or, in particular, in cancer cells. Published kinetic constants for *GCL* may also substantially differ for the enzyme present in JM-1 cells particularly under intracellular conditions.

## 4. Materials and Methods

### 4.1. Cell Culture

JM-1 rat hepatoma cells were obtained from George Michalopoulos (University of Pittsburgh, PA, USA) and were cultured in Dulbecco’s modified Eagles medium (DMEM, 3 g/L glucose, Mediatech, Manassas, VA, USA) supplemented with 10% fetal bovine serum (FBS) containing streptomycin and penicillin. For bioreactor experiments, cells were trypsinized, washed in phosphate-buffered saline (PBS) and electrostatically encapsulated.

For labeling experiments, [3,3′-^13^C_2_]-cystine was produced by dissolving [3-^13^C]-L-cysteine (Cambridge Isotope Laboratories, Tewksbury, MA, USA) in a minimum of distilled water, adjusting the pH to 8–8.5 with 0.5 M NaOH and stirring overnight while exposed to room air. The resulting solution/suspension of [3,3′-^13^C_2_]-cystine was added to cysteine- and methionine-deficient Dulbecco’s modified Eagles medium (DMEM, Mediatech, Manassas, VA) with unlabeled methionine and stirred to completely dissolve the labeled cystine to a concentration of 200 μM. An amount of 10% FBS was added to this solution with antibiotics to produce labeled growth media (^13^C-DMEM).

For the glutathione conjugation studies, 140 μL of 0.21 M monobromobimane (mBBr, EMD Biosciences, La Jolla, CA, USA) in DMSO was added to 100 mL of ^13^C-DMEM media. This solution was used immediately after the mBBr dissolution.

### 4.2. Encapsulation Methods

Cells were encapsulated using an electrostatic bead generation apparatus described previously [33] with modifications [13]. Briefly, JM-1 cells at a density of 0.5–1 × 10^9^ cells/mL were suspended in 2% sodium alginate (Sigma-Aldrich, Inc., St. Louis, MO, USA) solution at a 1:1 ratio. The resulting alginate–cell suspension was placed in a 1 cc syringe fitted with a 24-gauge angiocatheter. The angiocatheter was pierced at the hub with a 27-gauge needle that served as the positive electrode and a second grounded electrode was immersed in the CaCl_2_ receiving bath for the electrostatic casting process. A syringe pump was used at flow rates between 0.75 to 1.5 mL/min in the presence of the high electrostatic potential (ca. 6 kV) to produce droplets that polymerized into solid calcium alginate beads (diameter 500 μm) containing encapsulated JM-1 cells in the receiving bath. Encapsulates were transferred to the culture media within two minutes to avoid excessive exposure to calcium.

### 4.3. ^31^P and ^13^C NMR Spectroscopy

The ^31^P and ^13^C NMR experiments of the JM-1 cells were performed on a narrow-bore 14.1 T Varian INOVA spectrometer equipped with a 10 mm broadband probe (Venus Probes, Livermore, CA, USA). The receiver frequency of the probe was tuned to ^31^P at 242.78 MHz and ^13^C at 150.92 MHz. The ^31^P time courses at 15 min were acquired before and after the drug treatment using a repetition time (TR) = 2 s, number of transients (nt) = 334 and a 77° flip angle. The ^31^P spectra were zero filled to 40,000 points and line broadened 15 Hz using Gaussian–Lorentzian apodization. Then, ^13^C time courses at 30 min intervals were acquired, during the initial uptake of the ^13^C-cysteine isotopic label, using a TR = 2 s, nt = 600 and a 60° flip angle, and ^13^C time courses at 1.2 min temporal resolution were acquired during the drug treatment using a TR = 2 s, nt = 24 and a 60° flip angle. NMR data was processed off-line with ACD/Specmanager software (ACD/Labs, Toronto, Ontario, Canada). The ^31^P metabolites were identified using α-NTP (−7.5 ppm) as an internal reference. The area beneath the ^13^C resonances from glutathione, glutathione-bimane conjugate, were obtained by integrating the signals at 25.8 and 33.6 ppm, respectively. These experimentally acquired integrated intensities were used in fitting the exponential curves. NMR integrated intensities of the ^13^C were used to calculate metabolite concentrations in the NMR tube by acquiring data from concentration standards under equivalent acquisition parameters.

### 4.4. Cell Perfusion and Treatment

Alginate-encapsulated JM-1 cells were perfused at 3 mL/min with unlabeled DMEM/10% FBS in an NMR-compatible bioreactor [34] that was modified into a fluidized-bed configuration [13]. An initial ^31^P NMR spectrum was acquired to determine the viability of the cells. The perfusate was then switched to ^13^C-DMEM followed by serial ^13^C spectral acquisitions in 30 min intervals over 17 h. Another ^31^P NMR spectrum was acquired, prior to the dosing treatment to ensure the cells were still viable. Just prior to dosing, the ^13^C NMR spectra were acquired serially in 1.2 min time intervals for JM-1. The perfusate was switched to the mBBr-containing ^13^C-DMEM media for 10 min then switched back to mBBr-free ^13^C-DMEM. The ^13^C NMR spectral acquisitions continued for up to 6 h total acquisition of the time course with 1.2 min intervals. Afterwards, a final ^31^P NMR spectrum was acquired to determine the effects of the drug treatment.

### 4.5. Metabolite Isolation

Upon termination of the NMR experiments, alginate beads were transferred to Eppendorf centrifuge tubes and rinsed twice with a cold saline solution. After which, the beads were dissolved with 100 mM citrate in PBS, centrifuged for 2 min to separate the pellet then rinsed with PBS and the dissolved alginate was aspirated off. Metabolites from the rat hepatoma cell line were extracted with a modified perchloric acid extraction procedure [35].

### 4.6. Metabolite Assays

A Waters Acquity Ultra Performance Liquid Chromatography (UPLC) system equipped with a photodiode array UV detector (Waters, Milford, MA, USA) was used for the assay of the bimane conjugates of cysteine, γ-glutamylcysteine (γ-glu-cys), cysteinyl-glycine (cys-gly) and glutathione from the mBBr-treated cell extracts. The column was a Waters symmetry ethylene bridged hybrid (BEH) C18-column (2.1 × 100 mm i.d., 1.7 µm). The detected wavelength was set at 390 nm.

### 4.7. Evaluation Mercapturate Pathways

JM-1 cells in monolayer were treated with mBBr and the extracellular medium was sampled at 10, 30 and 90 min time points. The media was analyzed for the presence of N-acetyl-L-cysteinyl-bimane by UPLC.

### 4.8. Mass Spectrometry

The fractional enrichment of glutathione was determined by liquid chromatography/mass spectrometry of the glutathione-bimane conjugate. At the conclusion of the NMR experiment, the cells in beads were treated with mBBr, recovered and treated with perchloric acid as noted above. An Agilent 6500 Series Accurate-Mass Quadrupole Time-of-Flight (Q-TOF) LC/MS operating in positive ion mode was used to detect the isotope distribution of glutathione-bimane C_20_H_27_N_5_O_8_S (M+H = 498 amu) at *m/z* 498, 499, 500 to determine the fractional enrichment [5].

### 4.9. Data Fitting

Data for the initial label uptake into glutathione and its resynthesis after mBBr depletion was fit to an exponential function *y = A + B*(1 *− exp*(*−k × t*)) by least squares methods in Microsoft Excel.

## 5. Conclusions

We have been developing methods to best monitor glutathione metabolism in intact tissue. Tracing metabolism through the use of isotope labels is only part of the analysis. Relating these experimental data to flux through metabolic pathways require validated mathematical models of glutathione metabolism in the tissue. This model building requires the translation of known metabolic pathways into a series of differential equations and the use of experimentally determined kinetic constants and is currently underway in our laboratories. Future work also will now extend to other cell systems and stress inducers.

## Figures and Tables

**Figure 1 ijms-23-04733-f001:**
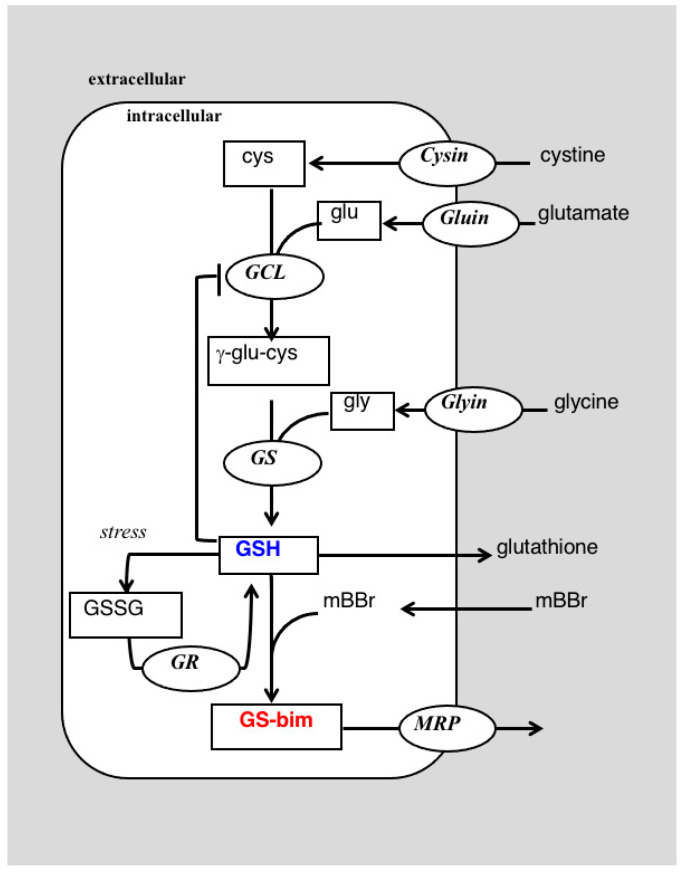
A portion of the glutathione metabolic network showing the reaction of cysteine (cys) and glutamate (glu) to form γ-glutamylcysteine (γ-glu-cys) in a reaction catalyzed by glutamyl cysteine ligase (*GCL*). Glutathione can act by feedback inhibition to slow *GCL* activity. The second step adds glycine (gly) in a reaction catalyzed by glutathione synthetase (*GS*) to form glutathione (GSH). Under oxidative stress GSH can be oxidized to its disulfide (GSSG) which can be reduced via glutathione reductase (*GR*) back to GSH. Glutathione can react spontaneously with monobromobimane (mBBr) or via an enzyme catalyzed reaction to form the glutathione-bimane conjugate (GS-bim) which may be extruded from the cell by the multi-drug resistance related protein (*MRP*).

**Figure 2 ijms-23-04733-f002:**
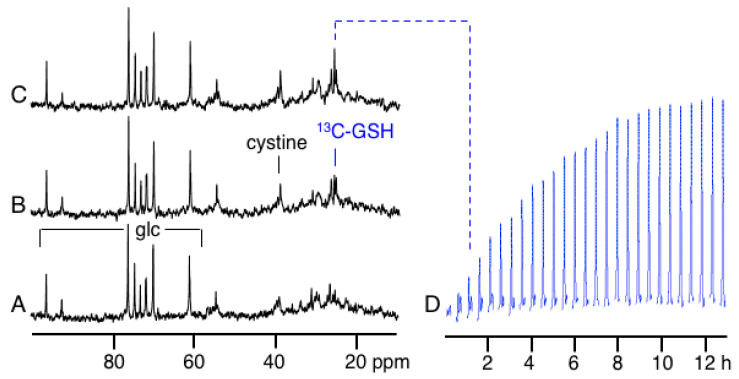
(**A**) A portion of the natural abundance ^13^C NMR spectrum of perfused JM-1 cells. The resonances due to glucose (glc) in the media are indicated. (**B**) The same cells 30 min after addition of 200 mM [3,3′-^13^C_2_]-cystine to the perfusate. The cystine resonance is detectable at 39.1 ppm and the resonance from the 3-position of the cysteinyl-residue of glutathione (^13^C-GSH) at 25.8 ppm. (**C**) The same cells, after an additional 30 min after switching to [3,3′-^13^C_2_]-cystine in the media. (**D**) A time-course of the region of each spectrum between 25.3-26.3 ppm showing the growth of the ^13^C-GSH resonance just prior to and after the addition of [3,3′-^13^C_2_]-cystine with spectra obtained every 30 min.

**Figure 3 ijms-23-04733-f003:**
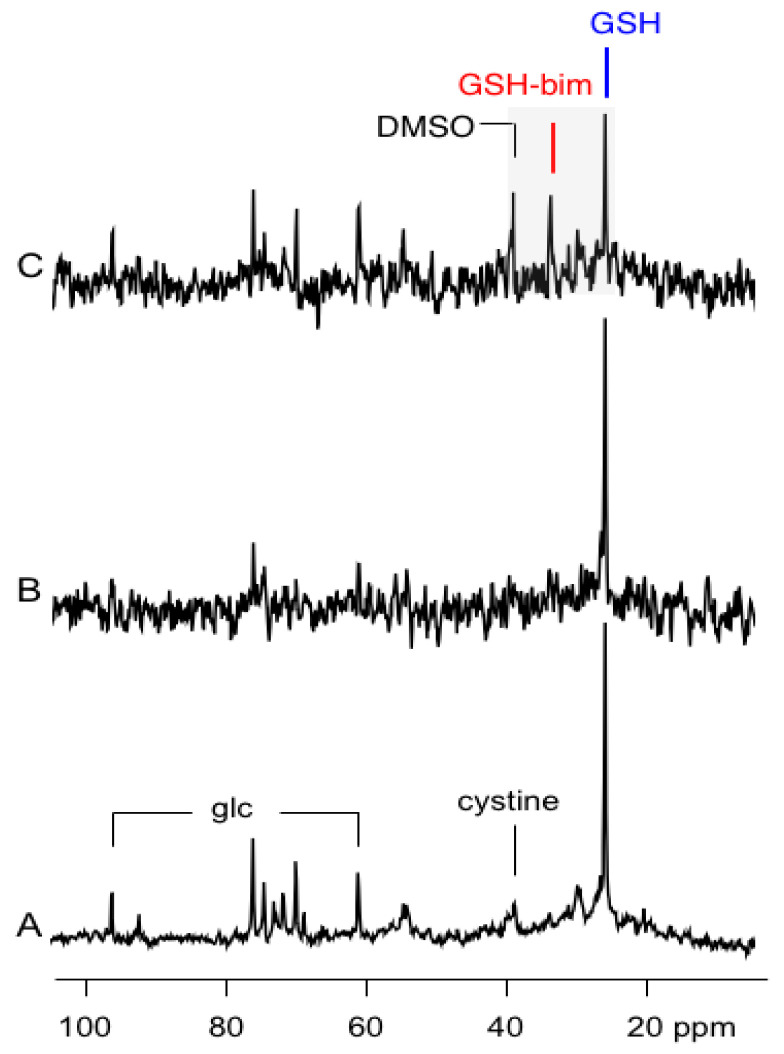
(**A**) A portion of the ^13^C NMR spectrum of perfused JM-1 cells 13.5 h after the switch over to [3,3′-^13^C_2_]-cystine. The data is obtained after a 30 min scan. The resonances due to glucose (glc) and cystine in the media are indicated. (**B**) The same cells after 13.5 h after switch over to [3,3′-^13^C_2_]-cystine obtained after a 1.2 min scan. (**C**) The same cells 9.6 min after adding a solution of mBBr in DMSO to the perfusate. The shaded region shows the three resonances of interest between 25–40 ppm of the ^13^C-spectrum.

**Figure 4 ijms-23-04733-f004:**
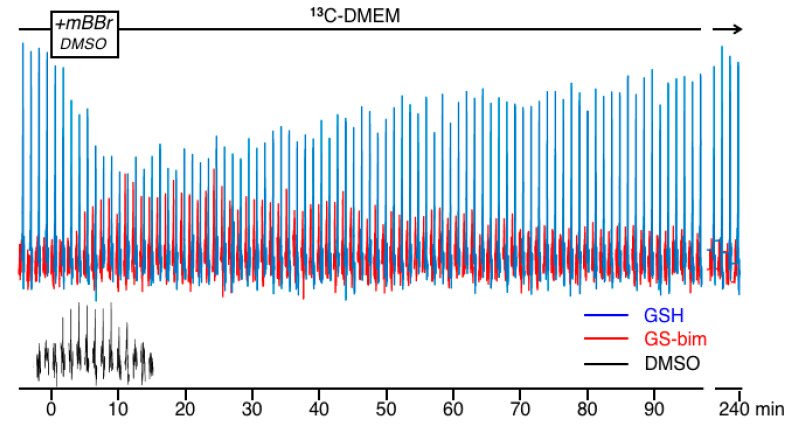
Time course of the ^13^C data collected in 1.2 min time intervals showing the region between 25–35 ppm just prior to, during and after exposure to media containing mBBr in DMSO. The box in the timeline above the spectra shows the time of exposure to mBBr. The resonances in blue are that for ^13^C-GSH at 25.8 ppm. The resonance in red is for the glutathione conjugate GS-bim detected at 33.6 ppm. For clarity, the resonances from the DMSO vehicle are plotted in black in the same time frame at the bottom of the figure.

**Figure 5 ijms-23-04733-f005:**
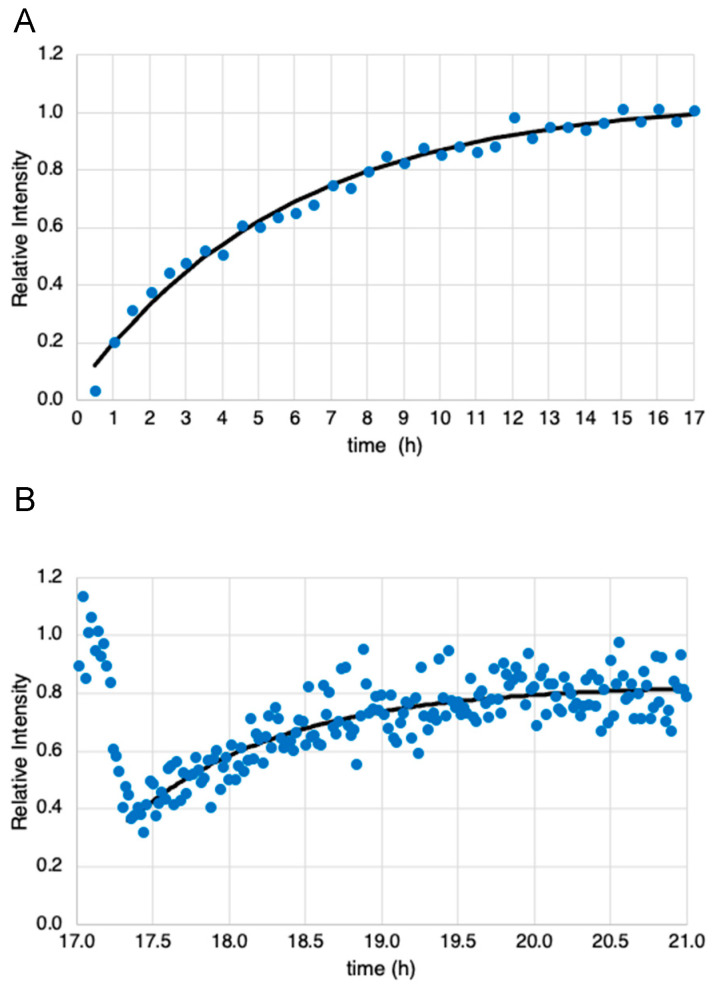
(**A**) Plot of the time course of the integrated intensity of ^13^C-GSH resonance collected in 30 min time intervals after switching to growth media containing [3,3′-^13^C_2_]-cystine. This was obtained from data similar to that in Figure 2D. The black line shows an exponential fit to the curve with rate constant *k* = 0.172 h^−1^. (**B**) Plot of the time course of the integrated intensity of the ^13^C-GSH resonance collected in 1.2 min time intervals just prior to, during and after the addition of mBBr to the perfusate. This was obtained from data similar to that in Figure 4. The black line shows an exponential fit to the portion of the data that represents glutathione resynthesis with a rate constant *k* = 1.04 h^−1^.

**Table 1 ijms-23-04733-t001:** Metabolite levels in JM-1 cells determined by UPLC.

Metabolite	Fmol/Cell
Glutathione	6.8 ± 20.5
Cysteine	4.46 ±1.63
γ-Glu-cys	1.11 ± 1.92
Cys-gly	4.74 ± 1.59

## Data Availability

All data generated and analyzed are included in the published article.

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
