# Peer review of "Direct Detection of Glutathione Biosynthesis, Conjugation, Depletion and Recovery in Intact Hepatoma Cells"

_ijms, 2022, doi:10.3390/ijms23094733_

Round 1

Reviewer 1 Report

The work is related to the development of approaches for direct detection of glutathione biosynthesis and its inactivation in JM-1 intact hepatoma cells. In particular, the principal possibility of direct detection of this tripeptide after the addition of a labeled precursor 200 mM [3.3’-13C2]-cystine was demonstrated using NMR spectroscopy. A decrease in the content of labeled glutathione was also recorded with the addition of thiol alkylating agent monobromobimane (mBBr). The work was done at a good methodological level. The data obtained is not in doubt. The authors chose a highly sensitive method for NMR detection of 13C-labeled compounds, which allowed them to make subtle observations.

The authors were able to demonstrate the dynamics of an increase in the content of labeled glutathione in response to the addition of a labeled precursor, and then, changes in the content of labeled glutathione upon the addition of a glutathione-binding agent. In this regard, it would be interesting to further use the existing methodological base also for the detection of the “work” of glutathione (change in its content) in response to glutathione-consuming effects, such as protecting cells from oxidative stress through reduction of reactive oxygen/nitrogen species or by inactivating toxic electrophilic xenobiotics. In this case, it would be possible to quantify the effect of exposure to glutathione homeostasis.

When interpreting the obtained data, it is not completely clear why a significant increase in glutathione occurs in the cell after the addition of the [3.3’-13C2]-cystine precursor for a long time in JM-1 cells (Fig. 2D)? Is the addition of a precursor in the amount of 200 mM a kind of stress for the cell leading to an increase in glutathione content? Or can the precursor be consumed otherwise than stored in the form of glutathione tripeptide? Why cannot the exogenously introduced precursor be consumed otherwise than stored in the form of glutathione tripeptide, and is not involved in other metabolic pathways of sulfur-containing amino acids in JM-1 cells?

It is also not entirely clear why the dynamics of glutathione depletion is not traced over time when the alkylating agent mBBr is added, for example, after 240 minutes (Figure 4). This is due to the fact that the studied concentration of mBBr was not sufficient to bring the content of glutathione to its original state (Fig. 2a), and it was impossible to increase the concentration of mBBr due to toxicity?

Comments:
Table 1.

The data presented in Table 1 were obtained in this study or have been obtained previously. If this is data from a previous work, then give a link to the source. If these are the data of this work, indicate in the section Materials and Methods by which method they were obtained (in particular, the concentration of Cys-gly)

Figure 3

It would be clearer if in figure 3 add more than one time point after adding a solution of mBBr in DMSO to the perfusate. The authors give only 9.6 min after adding (Fig 3C), and the dynamics of the process is shown in Figure 4. Add a time point from the interval of 60-240 minutes (after introducing mBBr) in order to present in one figure the entirety of the events (switch over to [3,3’-13C2]-cystine, after some minutes mBBr introduction, and what happens after few hours).

Author Response

We thank reviewer #1 for their comments. We have made some changes to the text in light of these comments and believe that this improves the clarity of our presentation. The detailed responses to the reviewers’ comments are listed below:

Response to Reviewer #1:

Comment: “…it is not completely clear why a significant increase in glutathione occurs after the addition of [3,3’-13C2]-cystine…(Fig 2D)?”

Reply: There is no increase in glutathione upon addition of labeled cystine. The glutathione levels remain unchanged as the media component is switched from unlabeled cystine to [3,3’-13C2]-cystine. Fig 2D shows the increase in the fraction of glutathione that is labeled with the 13C isotope as the glutathione is now synthesized from labeled cystine in unstressed cells. We have added a sentence (line 114) to clarify this point.

Comment: “Is the addition of precursor in the amount of 200 mm a kind of stress for the cell leading to an increase in glutathione content?”

Reply: In these experiments, the cell growth media normally contains unlabeled 200 mm cystine. In these experiments, the unlabeled cystine is replaced with the exact same concentration of [3,3’-13C2]-cystine, so there is no stress imparted on the cells.

Comment: “…can the precursor be consumed otherwise…’

Reply: Yes, the [3,3’-13C2]-cystine may undergo metabolism through any number of additional pathways in addition to its transformation into glutathione. However, glutathione is the highest concentration of any molecule which contains cysteine so it is easily detectable in our experiments. In these experiments, we do not detect 13C in other cell components likely because of their low concentration. This is a good point and we have now added a sentence (line 116) to note this.

Comment: “…why the dynamics of glutathione depletion is not traced over time … for example, after 240 minutes (Figure 4).  This…was not sufficient to bring the content of glutathione to its original state (Fig. 2A)…?”

Reply: The dynamics shown in Figure 4 is essentially a complete time-course of the dynamics of glutathione depletion. The ‘original state’ prior to the addition of mBBr is not shown in Fig. 2A. The pre-dosing (i.e. ‘original state’ of the glutathione) is shown in the furthest left spectra in Figure 4, just prior to mBBr addition. From these data, you can see that the glutathione returns to its original state after 240 min. This is noted in the text (page 6, line 159) as ‘the 13C-GSH resonance has recovered to nearly pre-dosing levels…’

The data in Fig. 2A shows the spectrum from cells prior to 13C-labeling, when the glutathione is not easily detected by NMR. The data in Figure 4 was obtained with cells exposed to 13C-DMEM (as noted at the top of this figure) so a return to the pre-labeled state in Fig. 2A would not be possible unless the media was switched back to unlabeled cystine.

Comment: “The data presented in Table 1 were obtained in this study or been obtained previously.”

Reply: The Methods section notes the UPLC assay procedures for the metabolites contained in Table 1 and now includes ‘cysteinyl-glycine’ as a component that was assayed (line 399) and the Table Heading now refers to the UPLC assay (line 110).

Comment: Clarity of Fig. 3. “…present in one figure the entirety of the events…”

Reply: The NMR is not sensitive enough to detect mBBr itself but only the DMSO vehicle. As the reviewer noted, the dynamics of the process is shown in Figure 4. The DMSO signal is not detected after 15 min so that data is not presented.

As for presenting one figure with all events, the number of spectra and difference in time intervals make that difficult. The switch over to [3,3’-13C2]-cystine acquired data every 30 min, whereas mBBr treatment acquired data every 1.2 min. Combining everything into one figure would result in a very complex figure which would be difficult to interpret at a size that would fit within margins of the journal.  

Reviewer 2 Report

The manuscript presents a new interesting tool for studies of glutathione metabolism, based on the application 13C-NMR after labeling cells with [3,3’-13C2]-cystine, with a time resolution below 2 min This approach allowed for estimation of the kinetics of glutathione depletion and resynthesis after depletion with an exogenous electrophile. This approach may be valuable in studies of glutathione metabolism.

Remarks:

Line 83: µm?

Table 1, please indicate by which method the data were obtained

Conclusions: It is rather unusual to cite other studies in the conclusions (first sentence) which should summarize the presented study.

Author Response

We thank reviewer #2 for their comments. We have made some changes to the text in light of these comments and believe that this improves the clarity of our presentation. The detailed responses to the reviewers’ comments are listed below:

Response to Reviewer #2:

Comment: “Line 83: mm?”

Reply: The typographical error has corrected to indicate mm as assumed by this reviewer.

Comment: “Table 1”

Reply: The Table heading has been adjusted to indicate the method used (line 110)

Comment: Conclusions section

Reply: Good point, we have eliminated the sentence and reference to previous work to concentrate on the conclusions we can draw from this work (page 13).
